# Association between Fear of Falling and Seven Performance-Based Physical Function Measures in Older Adults: A Cross-Sectional Study

**DOI:** 10.3390/healthcare10061139

**Published:** 2022-06-19

**Authors:** Wen-Ni Wennie Huang, Hui-Fen Mao, Hsin-Min Lee, Wen-Chou Chi

**Affiliations:** 1Department of Physical Therapy, College of Medicine, I-Shou University, Kaohsiung 82445, Taiwan; hwennie@isu.edu.tw (W.-N.W.H.); hmlee@isu.edu.tw (H.-M.L.); 2School of Occupational Therapy, College of Medicine, National Taiwan University, Taipei 100025, Taiwan; huifen02@gmail.com; 3Department of Physical Medicine and Rehabilitation, National Taiwan University Hospital, Taipei 100225, Taiwan; 4Department of Occupational Therapy, College of Medical Science and Technology, Chung Shan Medical University, Taichung 40201, Taiwan; 5Occupational Therapy Room, Department of Physical Medicine and Rehabilitation, Chung Shan Medical University Hospital, Taichung 40201, Taiwan

**Keywords:** fear of falling, balance, physical function, older adults

## Abstract

Fear of falling (FOF), a common phenomenon among older adults, may result in adverse health consequences. The strength of the association between FOF and physical function among older adults has not been well compared in previous studies. Therefore, a cross-sectional study was performed on 105 older adults to determine and compare the strength of the association between FOF and seven common physical function measures. After controlling for age, logistic regression models were fitted for each physical function measure. According to odds ratios, the Berg Balance Scale (BBS), Short Physical Performance Battery, gait speed, and Timed Up & Go Test were associated with the identification of FOF. Based on a *c*-statistic value of 0.76, the BBS, a common and quick assessment of functional balance tasks, was found to be able to distinguish between fearful and non-fearful older adults. Interventions targeted to improve lower-extremity physical functions, especially functional balance ability, may help prevent or delay the adverse consequences of FOF.

## 1. Introduction

Fear of falling (FOF) is a common phenomenon experienced by community-dwelling older adults, with a reported prevalence of 21–85% [1,2]. In older adults, FOF is considered a major problem because it can lead to physical, psychological, functional, and social disturbances [3]. Fearful older adults may avoid or restrict activities in an attempt to avoid falls even though they are able to perform these activities. When present at an excessive level, the avoidance of activities may lead to functional decline [4]. The consequences of FOF include falls [2,5], depression [3], decreased ability to perform activities of daily living (ADL) [4], and activity restriction [3].

Several studies have identified the risk factors for FOF including non-modifiable and modifiable risk factors [1,3,6]. Non-modifiable risk factors include female sex, older age, and history of at least one fall [3], whereas modifiable risk factors include physical characteristics, such as gait and balance [7], and psychological characteristics, such as poor subjective health status and depression [6]. Among these, modifiable risk factors can be managed through interventions and lifestyle modifications to decrease the risk of FOF. Previous studies have identified the association between FOF and lower extremity physical function measures including gait speed [8,9], balance [10,11], lower-extremity muscle strength [8,12], and upper-extremity muscle strength [12,13]. Despite the findings about the associations between FOF and the physical function measures, the comparisons of the strength of these associations have not been well investigated. The association between FOF and upper-extremity function, such as fine motor dexterity, has not been identified. Understanding the strength of the associations between FOF and different measures of lower- and upper-extremity physical functions may help clinicians identify the domains of physical functions that are more relevant to FOF, thereby prioritizing the interventions targeted for FOF.

This study aimed to compare the performance of seven commonly used physical function measures between fearful and non-fearful older adults, and to examine and compare the association between FOF and the seven physical function measures including Berg Balance Scale (BBS), Short Physical Performance Battery (SPPB), gait speed, Timed Up & Go (TUG) Test, Five Times Sit to Stand Test (FTSST), grip strength test, and nine-hole peg test.

## 2. Materials and Methods

### 2.1. Design

Older adults were invited to join a single-session face-to-face interview at two senior living facilities, and their demographic and clinical characteristics, FOF status, experience with falls, and physical function data were collected by trained evaluators.

### 2.2. Participants

The inclusion criteria were as follows: age ≥ 65 years, ambulatory with or without a walking aid, and Mini-Mental State Examination (MMSE) score ≥24 [14]. The exclusion criteria were ambulatory requiring assistance from another person, non-ambulatory, and persistent lower-extremity pain-limiting ambulation. Two sets of older adults (45 and 60 participants) were recruited from two senior living facilities from February 2011 to January 2012. The study was approved by the Institutional Review Board for Human Subject Research of I-Shou University (ISU-IRB-99-003), and informed consent was obtained from all participants.

A sample size of 105 was used in this study. The sample size was determined to detect a significant clinical difference between fearful and non-fearful older adults, and to ensure sufficient case numbers for logistic regression. The sample size for group comparisons was estimated based on data from a clinical trial investigating FOF and gait speed performance in older adults [15]. A sample size of 32 subjects, 16 in each group, is sufficient to detect a clinically important difference of gait speed of 38.1 cm/s between fearful and non-fearful older adults assuming a standard deviation of 33.0 using a two-tailed t-test of difference between means with 90% power and 5% level of significance. The sample size for logistic regression was estimated based on the recommendation of Long [16], suggesting that the minimum number should be 100.

### 2.3. Measures

#### 2.3.1. Demographic and Clinical Characteristics

Demographic and clinical characteristics including cognitive function and depressive symptoms were evaluated. The MMSE [14], a 30-item interviewer-administered cognitive screening tool, was used to assess several dimensions of cognitive function including orientation, attention, memory, language, and visual-spatial skills; a score ≤23 is a generally accepted cut-off indicating the presence of cognitive impairment [17]. Older adults with a score ≤23 in MMSE were excluded from the study. Depressive symptoms were assessed using the 15-item Geriatric Depression Scale (GDS-15), with a score ≥5 indicating the presence of depression [18].

#### 2.3.2. FOF

FOF was evaluated using a single-item question: “Are you afraid of falling?” Responses were recorded as yes or no. Previous studies that used this single-item question [2,19,20] have shown good reliability [21], and correlation with the Falls Efficacy Scale-International [22] have been reported [23].

#### 2.3.3. Experience with Falls

A fall is defined as a sudden, unintentional change in position causing an individual to land at a lower level, on an object, the floor, or the ground as a consequence of a sudden onset of paralysis, epileptic seizure, or overwhelming external force [24]. History of falls in the previous year was evaluated using a single-item question: “Have you had a fall in the previous year?”

#### 2.3.4. Performance-Based Physical Function Measures

Seven performance-based physical function measures, namely, the Berg Balance Scale (BBS), Short Physical Performance Battery (SPPB), gait speed test, Timed Up & Go (TUG) Test, Five Times Sit to Stand Test (FTSST), grip strength test, and nine-hole peg test (NHPT), were completed during the interviews.

1.BBS

The BBS is a validated and commonly used measure of balance in older adults [25,26,27]. It consists of 14 functional balance tasks: sitting to standing, standing unsupported, sitting unsupported, standing to sitting, transfers, standing with eyes closed, standing with feet together, reaching forward with an outstretched arm, retrieving object from floor, turning to look behind, turning 360°, placing alternate foot on stool, standing with one foot in front of the other foot, and standing on one foot. The BBS has been shown to have excellent inter-rater (ICC = 0.98) and intra-rater (ICC = 0.99) reliability [26] and validity based on correlations determined using the Tinetti’s Performance-Oriented Mobility Index balance subscale (*r* = 0.91) and TUG test (*r* = −0.76) [26,27]. The BBS is a validated clinical tool for the prediction of recurrent falls [28], a future fall [29,30], and ADL disability [31].

2.SPPB

The SPPB is a clinical test that reflects general lower-extremity function and includes three tasks: balance, walking, and chair stand. A 4-point scale was evaluated for each task where the summary score ranged from 0 to 12 (12 being the best score) [32]. Satisfactory inter-rater reliability (ICC > 0.9) and test-retest reliability (ICC = 0.72) have been reported for SPPB [33]. The SPPB is a validated clinical tool for the prediction of future disability [31,32], mortality [34], and hospitalization in older adults [33].

3.Gait speed

Gait speed was measured using the SPPB testing, which was measured over a 4 m distance with a stopwatch and 4 m tape. Gait speed alone has been reported to be as good a predictor of ADL and mobility disability as the total SPPB summary score [32]. Gait speed measurements obtained from a 4 m walk test have been shown to have excellent test-retest reliability (ICC = 0.96–0.98) [35]. Gait speed is a validated tool for the prediction of hospitalization and functional decline [33].

4.TUG

The TUG was modified from an observational mobility scale [36]. The time required to stand up from a straight back chair with arms, walk 3 m at a usual speed, turn, return, and sit down again was recorded with a stopwatch [37]. High inter-rater (ICC = 0.99) and intra-rater (ICC = 0.99) reliability have been reported for the TUG [37]. Satisfactory construct validity for balance, mobility, and ADL was demonstrated by correlations with the BBS (*r* = −0.81), gait speed (*r* = −0.61), and Barthel Index (*r* = −0.78) [37]. The TUG is a risk factor for fall [38] and a validated tool for the prediction of ADL disability [31].

5.Five Times Sit to Stand Test

Five Times Sit to Stand Test was measured using the SPPB testing. Participants were seated in a chair and asked to stand and sit five times as quickly as possible with their arms crossed over their chests. FTSST was timed in seconds from the command to go until the participant straightened up completely for the fifth time. FTSST has significant correlations with lower-extremity strength [39] and balance [40], and it also predicts disability and falls [41].

6.Grip strength

Grip strength was measured with a Jamar^®^ hydraulic hand dynamometer (Asimov Engineering Co., Los Angeles, CA, USA) for the dominant hand. The participants were seated with forearm resting on a table, elbow bent, and wrist in a neutral position, as recommended by the American Society of Hand Therapists [42]. Satisfactory test reliability for left- and right-hand grip strength (0.84 and 0.81, respectively) has been reported in women aged 60–90 years [43]. Midlife grip strength has been shown to predict walking disability and self-care disability after 25 years [44].

7.NHPT

The NHPT was developed to measure fine manual dexterity. The NHPT was administered in the study to explore possible association between fine manual dexterity and FOF. Test administration involves the time required to place nine pegs in holes on a 5-inch square board and then remove them. The NHPT has high inter-rater reliability (*r* = 0.97 for right hand and *r* = 0.99 for left hand) and good test-retest reliability (*r* = 0.69) [45].

### 2.4. Statistical Analysis

All data analyses were performed using SAS^®^ version 9.2 (SAS Institute, Inc., Cary, NC, USA). Demographic and clinical characteristics and physical functions were compared between fearful and non-fearful older adults. Student’s t-test was performed to compare age, gait speed, TUG, FTSST, grip strength, and NHPT results between fearful and non-fearful older adults. The Mann–Whitney test was performed to compare GDS, MMSE, BBS, and SPPB scores between fearful and non-fearful older adults. The Chi-square test was performed to determine whether fall history and sex were related to FOF. The Benjamini–Hochberg procedure [46] was applied to control the false discovery rate at an alpha level of 5% for the multiple comparisons between FOF and the seven physical function measures.

To identify risk factors for FOF, logistic regression models were fitted for each performance-based physical function measure, with a binary FOF outcome as the dependent variable and the data of BBS, SPPB, gait speed, TUG, FTSST, grip strength, and NHPT as the main independent variables of interest. After controlling for age, odds ratios and 95% confidence intervals were estimated. The odds ratio is a scalar measure of association between a factor and an outcome [47], and it represents the change in the odds for any increase of 1 unit in the corresponding risk factor. To evaluate the discriminatory capacity of each physical function measure for FOF, the *c*-statistic, the area under the receiver operator characteristic (ROC) curve, was estimated. A *c*-statistic closer to 1 indicates that the model assigns higher probabilities (based on combinations of independent variables) to all observations with the event outcome, compared with the non-event observations [48]. According to Hosmer and Lemeshow [49], the area under the ROC curve between 0.7 and 0.8 (0.7 ≤ *c* < 0.8) is considered acceptable discrimination, between 0.8 and 0.9 (0.8 ≤ *c* < 0.9) an excellent discrimination, and >0.9 (*c* ≥ 0.9) an outstanding discrimination.

## 3. Results

In total, 105 older adults living independently or in assisted living at two senior living facilities volunteered to participate in this study (mean age, 81.2 ± 7.0 years; range, 66–96 years). Among the 105 participants, 61% reported FOF. Compared with the non-fearful participants, the fearful participants scored significantly lower in the BBS and SPPB, walked slower, and required longer time to complete TUG and FTSST (*p* < 0.05; Table 1). Comparisons of age, sex, GDS score, MMSE score, grip strength, and NHPT score revealed no significant differences between fearful and non-fearful older adults (*p* > 0.05; Table 1). Comparisons of demographic and clinical characteristics and performance-based physical function measures between the older adults recruited from two facilities revealed no significant differences (*p* > 0.05). A history of fall in the past year was more significantly common among the fearful older adults (*p* < 0.05; Table 1). The Benjamini–Hochberg procedure applied to adjust the *p*-values across the seven physical function measures showed that BBS, SPPB, gait speed, TUG time, and FTSST time remained significant between fearful and non-fearful older adults (Table 2).

The assumptions of logistic regression were tested and found satisfied. The collinearity statistics for each logistic regression showed no cause for concern regarding multicollinearity (VIF < 1.2). The casewise diagnostic showed standardized residuals less than 2.5 for all cases in the study, indicating that the data represented a fairly accurate model. Linearity of the independent variables with respect to the logit of the dependent variable was assessed via the Box–Tidwell procedure, and a Bonferroni correction was applied using the two terms in each logistic regression model. All independent variables were found to be linearly related to the logit of the dependent variable (*p* < 0.05).

After controlling for age, the odds ratios indicated significant associations between FOF and four lower-extremity function measures, i.e., the BBS, SPPB, gait speed, and TUG (*p* < 0.05; Table 3). The discriminatory capacity of the seven physical function measures was examined using the *c*-statistic (Table 3). The BBS had the highest *c*-statistic of 0.757 (95% CI: 0.61–0.90), followed by gait speed (*c*, 0.685), grip strength (*c*, 0.683), SPPB (*c*, 0.678), TUG (*c*, 0.674), FTSST (*c*, 0.637), and NHPT (*c*, 0.587). According to the criteria set by Hosmer and Lemeshow [49], the ability of the BBS to discriminate between fearful and non-fearful older adults reached the acceptable level.

## 4. Discussion

In this study, non-fearful older adults performed better regarding lower-extremity physical function measures including BBS, SPPB, gait speed, TUG test, and FTSST, but not in upper-extremity function measures including grip strength and NHPT. The strength of the associations between FOF and seven performance-based physical function measures was examined and compared. Functional balance measured by the BBS and walking assessed by SPPB, gait speed, and TUG were found to be risk factors for FOF (*p* < 0.05). According to the criteria set by Hosmer and Lemeshow [49], the BBS was the only physical function measure that achieved the acceptable discrimination of FOF (*c*, 0.76). The results of this study suggest that older adults with poorer functional balance and slower walking speed are at a greater risk of FOF, with functional balance being the domain of physical function that can discriminate between fearful and non-fearful older adults.

Functional balance measured by the BBS was found to be a risk factor for FOF and was able to discriminate between fearful and non-fearful older adults. The findings of the association between FOF and decreased balance ability in the study is consistent with those of previous studies [8,10,11,50]. Unlike previous studies that investigated the association between FOF and balance ability measured in laboratory-oriented settings [8,51], balance in the present study was measured using the BBS, a common functional balance measure that includes various balance tasks performed during daily activities. The results of the study suggest that functional balance is a significant risk factor for FOF, and its restoration may be essential for the management of FOF in fearful older adults. Compared with SPPB, gait speed, TUG test, and FTSST, the BBS yielded in the highest *c*-statistic, suggesting that it has a greater discriminatory capacity for FOF, and that functional balance is more related to FOF than other aspects of lower-extremity functions such as walking speed and lower-extremity strength. Fearful older adults in the study (fearful, 46.0 ± 8.8; non-fearful, 52.1 ± 3.2) may also be at a greater risk of future functional disabilities because a BBS score of <49.5 has been reported to be predictive of ADL disability at 12 months [31]. Therefore, interventions targeting functional balance activities may benefit older adults with FOF and could be considered a prioritized intervention strategy in the management of FOF.

The findings of the association between FOF and lower SPPB score [7,13,52], slower gait speed [8], longer TUG time [53], and longer FTSST time [54] are consistent with those of previous studies. In this study, the SPPB score was lower in fearful older adults (fearful, 6.7 ± 2.4; non-fearful, 8.4 ± 2.3), suggesting that fearful older adults were at a greater risk of functional disability and mortality, because a SPPB score of <7.4 is predictive of ADL disability at 12 months [31], and a SPPB score of <7 is predictive of hospitalization [33]. In this study, gait speed was slower in fearful older adults (fearful, 0.65 ± 0.23 m/s; non-fearful, 0.81 ± 0.26 m/s), suggesting that fearful older adults were at a greater risk of functional disability because a gait speed of <0.66 m/s is predictive for ADL disability at 12 months [31]. In this study, TUG time was longer in fearful older adults (fearful, 18.9 ± 7.8 s; non-fearful, 13.8 ± 3.4 s). Previous studies have found a TUG time of ≥12 s to be a risk factor for falling [38] and a TUG time of >13 s to be a risk factor for ADL disability at 12 months [31]. Both fearful and non-fearful older adults had a mean TUG time of >13 s in this study. Therefore, the implications of a greater risk of future falls and functional disability cannot be established. This is supported by a systematic review that has reported the limited ability of the TUG test to predict falls in community-dwelling elderly [55]. In this study, FTSST time was longer in fearful older adults (fearful, 21.0 ± 11.5 s; non-fearful, 16.0 ± 5.5 s). A cut-off FTSST time of >15 s has been identified for greater recurrent fall risk in community-dwelling older adults [56]. Therefore, the implications of a greater risk of recurrent falls cannot be established. Fearful older adults in the study may be at a greater risk of future functional disability because of the lower SPPB score and slower gait speed. Therefore, interventions targeting standing balance and walking may be beneficial for older adults with FOF.

The ability to grip and manipulate objects is considered the most important function of the hand. In this study, grip strength was different between fearful and non-fearful older adults (fearful, 14.2 ± 6.0 kg; non-fearful, 17.7 ± 7.3 kg; *p* < 0.05), but it was not identified by logistic regression analysis as a risk factor for FOF (*p* > 0.05). Deshpande et al. [12] reported similar results in which grip strength was different between fearful and non-fearful older adults but was not a risk factor for FOF in multiple linear regression analysis. Regarding manual dexterity, no significant differences in NHPT score was found between fearful (24.3 ± 6.2 s) and non-fearful (24.4 ± 7.5 s) older adults in this study (*p* > 0.05). Due to limited literature and lack of longitudinal studies, the relationship between upper-extremity physical function and FOF remain unclear. Based on the findings of the current study, the impact of FOF was greater on lower-extremity functions than on upper-extremity functions.

Depression was identified in previous studies as a risk factor for FOF [10,20,57]. However, the mean GDS scores of fearful (3.2 ± 3.1) and non-fearful (3.2 ± 3.4) older adults were not different in the present study (*p* > 0.05). Comparisons of proportions of older adults who scored ≥5 in the GDS using Fisher’s exact test revealed no differences between fearful (12.5%, 8/64) and non-fearful (9.8%, 4/41) older adults. The small number of older adults with depression may have resulted in this insignificant finding.

There are several limitations of the study. The current investigation only included older adults who volunteered to participate. The participants were possibly more aware of their personal health and more concerned about falling. The self-reported fall history might result in a memory bias. The data must be interpreted with caution because the data were collected 10 years ago and only older adults in two senior living facilities were examined. Future studies may explore the predictive values of functional balance ability for the development of FOF in older adults. It would be clinically useful to assess the effects of interventions targeting functional balance, standing balance, and walking in fearful older adults.

## 5. Conclusions

The current study extended and elaborated on previous work on physical function-based risk factors for FOF. Comparisons of the association between FOF and seven commonly used performance-based physical function measures suggested that lower-extremity physical function measures including BBS, SPPB, gait speed, and TUG are risk factors for FOF, and functional balance ability measured by BBS has the highest discriminatory capacity to distinguish fearful from non-fearful older adults. Interventions designed to improve lower-extremity physical functions, especially the functional balance ability, are recommended for older adults with FOF.

## Figures and Tables

**Table 1 healthcare-10-01139-t001:** Demographic and clinical characteristics of the study participants and physical function measures.

Characteristics Mean (SD)	All *n* = 105	Fearful *n* = 64	Non-Fearful *n* = 41
Demographic			
Age, years	81.20 (6.97)	81.32 (7.21)	81.03 (6.68)
Male, *n* (%)	34 (32.38)	18 (28.13)	16 (39.02)
Clinical			
Geriatric Depression Scale, 0–15	3.22 (3.16)	3.21 (3.06)	3.24 (3.38)
Mini-Mental State Examination, 0–30	26.52 (2.16)	26.28 (2.09)	26.88 (2.26)
History of fall in the previous year, *n* (%) *	27 (25.71)	21 (32.81)	6 (14.63)
Physical function measures			
Berg Balance Scale (0–56) *	48.56 (7.62)	46.00 (8.82)	52.11 (3.23)
Short Physical Performance Battery (0–12) *	7.36 (2.50)	6.72 (2.42)	8.35 (2.31)
Gait speed, cm/s *	71.09 (25.17)	64.71 (22.98)	80.81 (25.51)
Timed Up & Go Test, seconds *	16.82 (6.85)	18.90 (7.84)	13.76 (3.36)
Five Times Sit to Stand Test *	19.06 (9.88)	20.96 (11.49)	16.02 (5.47)
Grip strength, kg	15.50 (6.64)	14.17 (5.95)	17.68 (7.25)
Nine-hole peg test, seconds	24.33 (6.69)	24.27 (6.23)	24.43 (7.52)

* Student’s *t*-test, Chi-Square, or Mann–Whitney test, *p* < 0.05.

**Table 2 healthcare-10-01139-t002:** Benjamini–Hochberg correction for multiple testing.

Independent Variable	Dependent Variable	Original *p* Value	Benjamini–Hochberg *p* Value	Determination
gait speed	FOF	0.001	0.007	significant
SPPB	FOF	0.004	0.014	significant
BBS	FOF	0.003	0.021	significant
TUG	FOF	0.006	0.029	significant
FTSST	FOF	0.014	0.036	significant
Grip	FOF	0.062	0.043	not significant
NHPT	FOF	0.927	0.050	not significant

**Table 3 healthcare-10-01139-t003:** Association between FOF and the physical function measures: summary of logistic regression analysis.

Physical Function Measure	*c*-Statistic (95% CI)	Odds Ratio (95% CI)
Berg Balance Scale	0.757 (0.614, 0.900)	1.231 (1.033, 1.468) *
Short Physical Performance Battery	0.678 (0.563, 0.792)	1.321 (1.075, 1.624) †
Gait speed	0.685 (0.572, 0.798)	1.027 (1.007, 1.047) †
Timed Up & Go Test	0.674 (0.511, 0.839)	0.862 (0.749, 0.992) *
Five Times Sit to Stand Test	0.637 (0.514, 0.762)	0.943 (0.887, 1.003)
Grip strength	0.683 (0.534, 0.833)	1.094 (0.995, 1.204)
Nine-hole peg test	0.587 (0.429, 0.746)	1.021 (0.932, 1.117)

* *p* < 0.05; † *p* < 0.01.

## Data Availability

The data are not publicly available due to the privacy issues.

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
