# Peer review of "Association between Fear of Falling and Seven Performance-Based Physical Function Measures in Older Adults: A Cross-Sectional Study"

_healthcare, 2022, doi:10.3390/healthcare10061139_

Round 1

Reviewer 1 Report

Dear authors,

The manuscript is interesting only due to the association of variables. However, I do have many concerns about using the TUG test, as it was referred as a poor prospective predictor of falls according to a meta-analysis (https://pubmed.ncbi.nlm.nih.gov/24484314/).

See my other comments in the attached file.

Regards.

Reviewer 2 Report

Thank you for giving me the opportunity to review this paper. Please find my comments in the file.

Reviewer 3 Report

Dear authors:

The article explores a phenomenon relevant to the health of the elderly population - fear of falling.

There is congruence between the objective, method and results. The introduction is succinct and does not allow us to understand the state of the art on fear of falling in the elderly and the consequences for their health. The association of risk factors with fear of falling is made with studies older than 5 years, which is not necessary because there are several recent studies on the topic and systematic reviews that are not referenced.

In the subtopic participants you mention - two sets of older adults (45 and 60 59 participants) were recruited from two senior living facilities from February 2011 to January 2012.  

The data that you present in this study has 10 years old?

The method section benefited from an organisation according to the STROBE guidelines.

It's missing reference to ethical procedures and authorization of the study by an ethics committee.

The discussion draws on studies more than 10 years old, which limits the dialogue of the results with the most recent evidence.

Reference 48 is incomplete.

Round 2

Reviewer 1 Report

Thank you for your response.

Regards.

Author Response

Thank you again for the comments and feedback you provided.  We have learned and benefited a lot.

Reviewer 2 Report

Thank you for your revised version of the manuscript, it is much improved. The authors have responded well to most of the recommendations. However, the issue of sample size still needs to be addressed:

The first sentence in the aim is the comparison between non-fearful and fearful persons. Hence, a sample size calculation for this aim is necessary. The explanation in the cover letter of the estimation of sample size for the logistic regression model is very well explained and should be added to the manuscript. 

Author Response

Thank you for the suggestions.  The following paragraph has been added to 2.2 Participants to address the sample size issue. “A sample size of 105 was used in this study. The sample size was determined to detect a significant clinical difference between fearful and non-fearful older adults, and to ensure sufficient case numbers for logistic regression. The sample size for group comparisons was estimated based on data from a clinical trial investigating FOF and gait speed performance in older adults [1].  A sample size of 32 subjects, 16 in each group, is sufficient to detect a clinically important difference of gait speed of 38.1 cm/s between fearful and non-fearful older adults assuming a standard deviation of 33.0 using a two-tailed t test of difference between means with 90% power and 5% level of significance. The sample size for logistic regression was estimated based on the recommendation of Long [2], suggesting that the minimum number should be 100.”

Reference:

  1. Chamberlin, M.E.; Fulwider, B.D.; Sanders, S.L.; Medeiros, J.M. Does fear of falling influence spatial and temporal gait parameters in elderly persons beyond changes associated with normal aging? J Gerontol A Biol Sci Med Sci 2005, 60, 1163-1167, doi:10.1093/gerona/60.9.1163.
  2. Long, J.S. Regression Models for Categorical and Limited Dependent Variables; Sage: Thousand Oaks, CA, 1997.

Reviewer 3 Report

Dear authors:

I maintain the same concern, the data is more than 10 years old, as is the authorization of the ethics committee.

The article has the quality to be published, but the data are not current, which raises some questions.

The changes introduced improve the article.

Author Response

Thank you for the comments.  We do understand the data collected 10 years ago may raise some questions.  Therefore, a sentence has been added to the limitation section to address the issue.  “The data must be interpreted with caution because the data were collected 10 years ago and only older adults in two senior living facilities were examined.”